# Effect of Grafting on the Production, Physico-Chemical Characteristics and Nutritional Quality of Fruit from Pepper Landraces

**DOI:** 10.3390/antiox9060501

**Published:** 2020-06-08

**Authors:** Ramón Gisbert-Mullor, Costanza Ceccanti, Yaiza Gara Padilla, Salvador López-Galarza, Ángeles Calatayud, Giuseppe Conte, Lucia Guidi

**Affiliations:** 1Departamento de Producción Vegetal, CVER, Universitat Politècnica de València, Camí de Vera s/n, 46022 Valencia, Spain; ragismul@etsiamn.upv.es (R.G.-M.); slopez@upv.es (S.L.-G.); 2Department of Agriculture, Food and Environment, University of Pisa, Via del Borghetto 80, 56124 Pisa, Italy; c.ceccanti3@studenti.unipi.it (C.C.); giuseppe.conte@unipi.it (G.C.); lucia.guidi@unipi.it (L.G.); 3Instituto Valenciano de Investigaciones Agrarias, Centro de Citricultura y Producción Vegetal, Departamento de Horticultura, CV-315, Km 10,7, Moncada, 46113 Valencia, Spain; padilla_yai@gva.es; 4Interdepartmental Research Center Nutrafood “Nutraceuticals and Food for Health”, University of Pisa, 56124 Pisa, Italy

**Keywords:** antioxidant activity, ascorbic acid, *Capsicum annuum*, carotenoids, lycopene, phenolics, scion, VOCs

## Abstract

Grafting is a widely utilized agronomical technique to improve yield, disease resistance, and quality of fruit and vegetables. This work aims to assess the effect of grafting and fruit ripening on the production, physico-chemical characteristics, and nutritional quality of fruit from Spanish local pepper landraces. Landraces “Cuerno,” “Sueca,” and “Valencia” were used as scions, and “NIBER^®^” as the rootstock. Two ripening stages of the fruits were sampled: green and red. Grafting improved the yield and marketable quality and did not negatively influence the physico-chemical and nutritional characteristics of the fruit. It was noteworthy that the bioactive compound contents and antioxidant capacity were more related to maturity stage and genotype, and red fruit had a higher antioxidant capacity than green fruit. However, in all the scions, grafting significantly enhanced lycopene content in both red and green fruit. Another important effect of grafting was the volatile compound composition evidenced by discriminant analyses, which was characterized for the first time in the fruit of these landraces. The rootstock and scion combination could be a way to improve not only the production, but also the fruit quality of peppers.

## 1. Introduction

Sweet pepper (*Capsicum annuum* L.) is one of the most important vegetable crops grown in the world that covers 1.99 million hectares (ha) of crop-growing surface area [1]. It is of great cultural and economic importance because of its multiples uses and phenotypic diversity [2,3].

In the past century, modern breeding developed good-performance hybrid cultivars that are normally more productive, more resistant to disease and pests, and more uniform in germination, growth, and highest vigor terms [4,5]. This resulted in genetic erosion, the declining heterogeneity of the organoleptic characteristics of pepper fruits, and the replacement of using local pepper varieties [6,7]. Spain, similar to other countries in the Mediterranean Region, is characterized by the considerable versatility of agro-climatic regions, which promotes a wide range of different phenotypes [8]. Thus, Spain is considered a secondary center of diversity for peppers, especially for *C. annuum,* which was brought mainly from Mexico immediately after the discovery of America [9].

In addition, the genetic uniformity of cultivated pepper landraces enhances the incidence of biotic and abiotic stresses [6,8,10], with lower yields that contrast with the high yield needs to cover increased demand. From this point of view, grafting may provide an eco-friendly technology to increase yields in landraces and reinforce tolerance to biotic and abiotic stresses [10]. Based on its vigor, the rootstock has been utilized in both open field and protected cultivations [10], and grafting is used mainly with the Cucurbitaceae and Solanaceae families, which encompass the most important crops such as tomato, eggplant, cucumber, watermelon, melon, and pepper [11]. However, among these vegetables, grafting is a less common practice in peppers, probably because commercial rootstocks provide modest profits [12,13]. Several scientific reports have reported the effects of rootstock on pepper fruit quality, such as morphometric and yield characteristics [14,15], sweetness/acidity [16,17], or levels of functional compounds [18,19]. Another peculiarity of pepper is its aroma [20], which has become a quality parameter for consumers [21,22].

Evidently, quality is a set of attributes that determine consumer choice, and can be divided into the external fruit aspect, and also into nutritional and nutraceutical characteristics. These last properties have become increasingly important for consumers given the positive relation between fruits and vegetables and human health [23,24]. Many studies report how a healthy diet rich in fruit and vegetables (functional food) can help to delay senescence processes and also reduce the risk of pathologies including cancer and cardiovascular diseases [23]. The characteristic of functional food is its richness in antioxidant compounds and that it is capable of scavenging reactive oxygen species (ROS) that are the basis of cellular oxidative stress [25]. For this reason, consumers today demand fruits and vegetables with higher sensory and nutraceutical values that are sometimes more characteristic in landraces [26].

Sweet pepper fruit falls into the category of functional foods because it is rich not only in ascorbic acid, carotene, and phenols, but also capsaicinoids, xanthophylls, and flavonoids. It is thus characterized by its high antioxidant capacity [27,28]. The amount of those phytochemicals in peppers depends on many factors, including rootstock. However, opposite results about the influence of grafting on the amount of phytochemicals in pepper can be found in the literature [10,19,29,30]. 

To the best of our knowledge, no studies on the role and response of grafting on pepper landraces in fruit quality and productivity terms have been conducted. In this context, the present work focuses on the effect of grafting on pepper landraces through the characterization of the fruit productivity and quality of three typical sweet pepper landraces from Valencia (Spain) and by considering the effect of the fruit maturity stage as these landraces are consumed in both green and red fruit, depending on the final destination of production. To determine fruit quality, not only physico-chemical characteristics (dry matter, titratable acidity, color, and volatiles), but also nutraceutical aspects (phenols, vitamin C, pigments, and antioxidant capacity) were assayed to determine the “whole” fruit quality. The rootstock we utilized was “NIBER^®^,” an F1 hybrid that we obtained in a classic breeding program. It has been demonstrated as being tolerant to abiotic stress [31] with higher yields (range of 32–80%) compared to ungrafted plants or other commercial pepper rootstocks [32] tested in Mediterranean conditions. As scions, we utilized “Cuerno,” “Sueca,” and “Valencia,” three representative landraces of pepper morphologies in the Mediterranean Region from the Instituto Valenciano de Investigaciones Agrarias’ (IVIA, Spain) traditional germplasm collection. 

## 2. Materials and Methods

### 2.1. Plant Material

Three pepper landraces from Valencia (Spain), namely “Cuerno,” “Valencia,” and “Sueca,” were grafted onto rootstock F1 NIBER^®^, obtained from a collaboration agreement reached between the Universitat Politècnica de València (UPV, Spain) and the IVIA. The ungrafted “Cuerno” (CU), “Valencia” (VU), and “Sueca” (SU) plants were used as the control plants. These landraces were selected based on fruit morphology representing the three main typologies in landraces pepper shape in Valencia fruits from “Cuerno” are elongated with very slightly marked shoulders; “Valencia” fruits are blocky with four shoulders, “Sueca” fruits are triangular shaped with three locules. 

Seeds were sown on March 20, 2019, in 104-hole seed trays filled with enriched substrate for germination. After 2 months of sowing, plants were grafted by the tube-grafting method [33]. Two weeks after grafting, seedlings were transplanted in the field. The fruit from the ungrafted “Cuerno” plants (CU), “Valencia” (VU), “Sueca” plants (SU), and from the plants grafted onto NIBER, “Cuerno” (CG), “Valencia” (VG), and “Sueca” (SG) were used.

### 2.2. Soil-Field Experiment

The experiment was conducted from June to September in Moncada (Valencia, Spain; Latitude: 39.58951793357715, Longitude: −0.3955507278442383) in the IVIA’s experimental field. Soil was sandy clay loam (clay: 21.2%; silt: 11.8%; sand: 67%). Organic matter was 0.61%, pH 7.8, at 25 °C and EC 1:5 at 25 °C: 0.289 dS m^−1^. 

Plants were grown in the open air in single rows placed 150 cm apart with 40 cm between each plant. The experiment was laid out according to a complete randomized block design with three replicates. Each replicate consisted in 15 plants. Plant irrigation met 100% crop evapotranspiration (ETc), as described in Penella et al. [33] by means of a drip system. Nutrients were applied through the irrigation system at a rate (kg ha^−1^) of 200 N, 50 P_2_O_5_, 250 K_2_O, 110 CaO, and 35 MgO, as recommended by Maroto [34].

The average range of minimum and maximum temperatures during the experiment was 15–28 °C for June, 19–32 °C for July, 19–32 °C for August and 18–29 °C for September [35].

### 2.3. Fruit Yield and Quality Assessment

Fruits were harvested from 15 plants per replication from the end of July to mid-September in relation to fruit maturation. Quality commercial production was evaluated according to commercial practices, as were the measured fruits with physiological disorders, mainly blossom-end rot (BER).

Nine randomized fruits (3 per replication of plant material) were selected in July (green fruits) and mid-September (red fruits) to measure fruit quality, which included: physico-chemical characteristics (percentage dry weight, pulp thickness, color index determination, titratable acidity, volatile compounds) and nutraceutical characteristics (total phenol, lycopene, total chlorophyll, carotenoids and ascorbic acid content, antioxidant capacity).

### 2.4. Fruit Dry Material and Pulp Thickness

To determine the percentage of dry weight in green and red fruit, the fresh weight of samples (FW) was recorded. Dry weight (DW) was established after drying samples at 60 °C for 72 h in a laboratory oven. The percentage of DW (% DW) was defined as: (DW/FW) × 100. Pulp thickness was measured on three sides of each fruit in the equatorial area.

### 2.5. Fruit Color Index Determination

For the color index determination, the L (0–100, black to white), a (±red/green) and b (±yellow/blue) Hunter parameters of the color system were measured by a chromameter (Konica Minolta CM-700d). The color index was determined after making three determinations around the equatorial plane of fruit. Color intensity (Chroma) and hue angle (H) were calculated by the following equations: Chroma = ((a)^2^ + (b)^2^)^1/2^ and Hue angle = tg^−1^ (b/a) [36].

### 2.6. Titratable Acidity

Titratable acidity (TA) was determined by potentiometric titration with 0.1 M NaOH (Merck Co.) up to pH 8.1 using 10 mL of juice. Citric acid (Merck Co.) was used as a reference for the TA (% citric acid) calculations. The pHs of fruit juice were determined with a pH meter (HANNA HI 2212).

### 2.7. Total Phenolic Analysis and Antioxidant Capacity Measurements

The phenolic content was analyzed according to Dewanto et al. [37]. Briefly, 1 g of each sample was homogenized with a mortar in 4.0 mL 80% (v/v) methanol and centrifuged at 10,000 *g* for 15 min at 4 °C. Total phenolic content was determined by the Folin-Ciocalteau colorimetric method based on the procedure of Singleton and Rossi [38]. A 10 µL aliquot of the supernatant was mixed with 115 µL of distilled water, 125 µL of Folin-Ciocalteau reagent (Merck Co., Kenilworth, NJ, USA), and 1.25 mL of NaHCO_3_ (7%) (Sigma-Aldrich, Co., St. Louis, MO, USA). The absorption (Abs) of the solution was measured at 760 nm in a spectrophotometer (Ultrospec 2100 Pro, GE Healthcare Ltd., Chalfont St Giles, Buckinghamshire, UK). Each measurement was compared with a standard curve of gallic acid (Sigma-Aldrich, Co., St. Louis, MO, USA) and total phenols were expressed as mg gallic acid equivalent (GA) g^−1^ FW, on the basis of a standard calibration curve (Abs_760_ = 0.01146µg mL^−1^ – 0.01515; R^2^ = 0.932).

Antioxidant capacity was measured with the method reported by Brand–Williams et al. [39]. Briefly, 10 μL of phenolic extract were added to 990 μL of a solution containing 3.12 × 10^−5^ M of 2,2-diphenyl-1-picrylhydrazyl (DPPH, Sigma-Aldrich, Co., St. Louis, MO, USA) in methanol (99.9% purity). The drop in absorbance at 515 nm was measured against a blank solution (with no extract) after a 30-min reaction time at room temperature (optimized for the highest antioxidant concentrations in the extract) in a spectrophotometer (Ultrospec 2100 Pro). Each measurement was compared with a standard curve of trolox (Sigma-Aldrich, Co., St. Louis, MO, USA) solution and the results were expressed as mg trolox equivalent (TE) g^−1^ FW, on the basis of a standard calibration curve (Abs_515_ = 3.5887 µg mL^−1^ + 0.1412; R^2^ = 0.984). 

### 2.8. Ascorbic Acid Concentration

Ascorbic acid content was spectrophotometrically determined as described by Kampfenkel et al. [40]. Briefly, 0.3 g of each sample was homogenized in 1 mL 6% trichloroacetic acid (TCA, Sigma-Aldrich, Co., St. Louis, MO, USA) and centrifuged at 10,000 *g* for 10 min. The supernatant was immediately used to analyze ascorbate. The absorption of the solution was measured at 525 nm in a spectrophotometer (Ultrospec 2100 Pro). Ascorbic acid was expressed as mg g^−1^ FW, on the basis of a standard calibration curve (Abs_525_ = 0.07889 µg mL^−1^ + 0.0097; R^2^ = 0.994). 

### 2.9. Chlorophyll and Carotenoids Concentration

Chlorophyll (Chl) (a and b) and carotenoid (Car) content were determined spectrophotometrically as described by Porra et al. [41]. Briefly, 0.3 g of each sample was added to 1.5 mL acetone 80% (v/v) (Scharlab Co.) and centrifuged at 7000 *g* for 10 min. The supernatant was used for the analysis. The absorption of the solution was measured at 663 nm, 648 nm, and 470 nm in a spectrophotometer (Ultrospec 2100 Pro). Chlorophyll (a and b) and carotenoid content of the extracts was calculated by the following equations:Chl *a* = 12.25 × Abs_663_ − 2.55 × Abs_648_ (µg mL^−1^),(1)
Chl *b* = 20.31 × Abs_648_ − 4.91 × Abs_663_ (µg mL^−1^),(2)
Car = [(1000 × Abs_470_ – 1.82 × Chl *a*) − (85.02 × Chl *b*)]/198 (µg mL^−1^).(3)

Chlorophylls and carotenoids were expressed as µg g^−1^ FW.

### 2.10. Lycopene Concentration

The lycopene in pepper fruit was extracted using a hexane:ethanol:acetone (2:1:1; v:v:v) (Sigma-Aldrich Co., St. Louis, MO, USA) mixture following the method of Adejo et al. [42]. A powdered sample (0.001 g) was dissolved in 1 mL of distilled water and vortexed in a water bath at 30 °C for 1 h. Then 8.0 mL of hexane, ethanol, and acetone were added, capped, and vortexed again, followed by incubation in a dark cupboard for 60 min. Next, 1 mL of distilled water was added to each sample and revortexed before being left to stand and separate into phases. Care was taken to ensure that any formed bubbles had completely disappeared. The cuvette was rinsed with the upper layer of one of the blank samples before using more fresh blank samples to zero the spectrophotometer at 503 nm. The absorbance of the upper layers of the lycopene samples were read at the same wavelength of 503 nm in a spectrophotometer (Ultrospec 2100 Pro). The lycopene content of extracts was expressed as mg g^−1^ FW. Lycopene levels of the extracts were then calculated using:
Lycopene (mg kg^−1^ fresh wt) = (A_537_ × 8 × 0.55)/(0.10 × 172)
where 537 g mole^−1^ is the molecular weight of lycopene, 8 mL is the volume of mixed solvent, 0.55 is the volume ratio of the upper layer to the mixed solvents, 0.10 g is the weight of pepper fruit added, and 172 mM^−1^ is the extinction coefficient for lycopene in hexane.

### 2.11. Volatiles Organic Compound Analysis

The volatile organic compounds (VOCs) were determined by solid-phase microextration-gas chromatography-mass-spectrometry (SPME-GC/MS) [43]. VOCs were extracted from 2 g of finely minced fruit samples and put inside a 20-mL glass vial, closed with an aluminum cap provided with PTFE-septum. Samples were conditioned at 60 °C for 10 min. VOCs were collected using a divinylbenzene/carboxen/polydimethylsiloxane (DVB/Carboxen/PDMS) Stable Flex SPME fiber (50/30 µm; 2-cm long)/Supelco, Bellefonte, PA, USA). The SPME fiber was exposed to headspace for 30 min in a water bath at 60 °C. Before each analysis, the SPME fiber was conditioned for 30 min at 270 °C in the GC injector. The fiber was inserted into the injector of a single quadrupole GC/MS (TRAcE GC/MSm Thermo-Finningan, Waltham, MA, USA) set at 250 °C for 3 min in the splitless mode to keep the fiber in the injector for 30 min to achieve complete fiber desorption. The GC program conditions were the same as those reported by Povolo et al. [44]. GC was coupled with a Varian CP-WAX-52 capillary column (60 m × 0.32 mm; coating thickness 0.5 µm). The transfer-line and ion source were both set at 250 °C. The filament emission current was 70 eV. A mass range from 35 to 270 m/z was scanned at a rate of 1.6 amu/s. Acquisition was carried out by electron impact in the Full Scan (TIC) mode, and three replicates were run per fruit sample. VOCs were identified in three different ways: comparing with the mass spectra of the Wiley library (version 11/2008); injecting the authentic standard (2-methyl-3-heptanone); calculating LRI and matching with the reported indices [44,45]. Data were expressed as the relative abundance (%) of total VOCs.

### 2.12. Statistical Analysis

The results for all parameters, except VOCs, were subjected to a two-way ANOVA analysis using Statgraphics Centurion XVII (Statistical Graphics Corporation 2014) with treatment. Landraces were employed as factors of the analyses. Each ripening (green and red fruit) was separately analyzed. The percentage data were arcsin-transformed before analyzing. The least significant difference (LSD) at a 0.05-probability level was used as the mean separation test.

For VOCs, the statistical analysis was performed by the JMP software (SAS Institute Inc., Cary, NC, USA). The determination of ripening and the landrace effect on VOC composition was made by the following linear model:y_ijk_ = µ + R_i_ + V_j_ + R_i_ × V_j_ e_ijk_,
where y_ijz_ = dependent variables; R_i_ = fixed effect of the *i*th ripening level (green; red); V_j_ = fixed effect of the *j*th pen (2 levels); ε_ijk_ = random residual.

To comply with ANOVA assumptions, data were tested for normality by the Anderson–Darling test (*p* < 0.05).

Multiple comparisons among treatments were performed by Fisher’s least significant difference (LSD) at *p* ≤ 0.05.

A multivariate statistical analysis was performed to analyze the VOCs profile by three complementary techniques to discriminate six groups (CG, VG, SG, CU, VU, SU): stepwise discriminant analysis (SDA); canonical discriminant analysis (CDA); discriminant analysis (DA) [46]. Analyses were run separately for the green and red fruit because the significant difference in the VOC profile between both two ripening levels would have made the grafting effect less evident.

The minimum number of VOCs able to discriminate the six groups was obtained by the SDA, a statistical technique specifically considered to select the number of variables that better separate groups. Subsequently, CDA derives a set of new variables, called canonical functions (CAN), which are linear combinations of the original interval variables, as reported in the following equation:CAN = d_1_X_1_ + d_2_X_2_ + ... + d_n_X_n_,
where d_i_ are the canonical coefficients (CC) that indicate the contribution of each variable in composing CAN, and X_i_ are the scores of the n original variables.

CAN summarizes between-groups variation by highlighting their differences. In general, if k groups are involved in the study, k−1 CAN is extracted. Efficient separation between groups was calculated by the Mahalanobis distance and the corresponding Hotelling’s T-square test [47]. The Hotelling’s T-square test extends the Student’s *t*-test to the multivariate domain [48].

The ability of CAN to assign each sample to the six groups was calculated as the percent of correct assignment using DA [49]. The centroids of the six groups are calculated and, for each sample, the distances from the six centroids are evaluated. One sample is assigned to one of the six groups based on the shortest distance from the six groups’ centroids [49].

## 3. Results

### 3.1. Fruit Yield

Grafting had a significant effect on total and marketable yields compared to the values recorded in the ungrafted plants for all landraces (Figure 1A). The higher fruit yield was attributable to a significant increase in the number of fruits per plant, which significantly increased because of the grafting on all the landraces (Figure 1B). In addition, grafting decreased the amount of BER fruit (Figure 1C), parameters for which the grafting and landrace interaction was also significant. Non-marketable yield decreased in all the grafted landraces compared to the ungrafted landraces (−27% in “Cuerno,” −61% in “Sueca,” and −46% in “Valencia”—data not shown), as did the yield of the BER fruit (−28% in “Cuerno,” −71% in both “Sueca” and “Valencia”).

### 3.2. Fruit Physico-Chemical Characteristics

The percentage of DW was not affected by either grafted fruits /or landraces (Table 1). No differences in the grafting technique were found for the parameters H and C of the green fruit, whereas the H parameter significantly changed in relation to grafting or landrace in the red fruit, with no interaction between the two variability factors (Table 1). In particular, the H parameter was higher in the red fruit from the ungrafted plants and was significantly higher in the fruit from Valencia vs. “Cuerno” and “Sueca.” Moreover, no significant interaction between grafting and landrace was found in the red fruit for these parameters.

As a general rule, grafted plants produce green and red fruit with significantly higher titratable acidity than ungrafted plants (Figure 2A,B). Titratable acidity was similar in the green fruit produced by the three landraces, while this parameter was significantly higher in the red fruit produced by “Valencia” and “Sueca” compared to “Cuerno.” No interaction was found for both the green and red fruit between grafting and genotype (Figure 2A,B). It was noteworthy that titratable acidity was higher in the red fruit than in the green fruit, and pH values were lower (*p* ≤ 0.05; Student’s *t*-test). In the green fruit, pH values were lower in those produced by the grafted than the ungrafted plants (Figure 2C), and significant differences appeared among landraces. In the green fruit produced by the “Sueca” landrace, pH values were lower than those from “Cuerno” and “Valencia.” In the red fruit, lower pH values were recorded in the fruits from “Valencia” (Figure 2C,D). For pH values, a significant grafting and landrace interaction was found only for the green fruit. The lowest pH value was detected in the green fruit from the grafted “Valencia” plants, while the higher value went to the fruits from the same ungrafted landrace and the ungrafted “Cuerno” landrace (Figure 2C).

### 3.3. Nutraceutical Compounds and Antioxidant Capacity

Among the phytochemical compounds found in pepper fruit, phenols are of particular interest for their ability to scavenge free radicals. Independently of grafting and landrace, phenols concentration was significantly higher in the red than the green fruit (*p* ≤ 0.05; Student’s *t*-test) and it changed differently in both red and green fruit (Figure 3A). Indeed, grafting had a negative effect on phenols concentration in both colored fruit. In the green ones, and among landraces, those produced by “Valencia” had the highest value of these compounds (Figure 3B). In the red fruit, the landrace factor was not significant (Figure 3). However, the grafting and landrace interaction was significant and evidenced some differences: higher phenols concentration values for the fruit from all the ungrafted landraces and the grafted “Valencia” (Figure 3B).

Figure 3B,D show the ascorbic acid content in both the green and red fruit of the grafted and ungrafted pepper landraces. In the green fruit, grafting and landrace, and their interaction, were significant. Even for this antioxidant metabolite, it was evident in the green fruit that grafting enhanced content, which was not observed in the red fruit (Figure 3C,D). In both green and red fruit, ascorbic acid content was higher in the fruits of “Sueca” and “Valencia” and lower in those of “Cuerno.” For the green fruit, the landrace that produced fruit with the highest ascorbic acid content was the grafted “Valencia,” landrace, in which grafting enhanced ascorbic acid content, and also in the other landraces (Figure 3C).

Conversely, grafting negatively impacted the ascorbic acid content in the red fruit. Between landraces “Sueca” and “Valencia,” once again fruits were produced with higher ascorbic acid content (Figure 3D). No significant interaction between variability factors was found for ascorbic acid in the red fruit.

Antioxidant capacity was determined by the DPPH assay and was higher in the red fruit compared to the green ones (*p* ≤ 0.001, Student’s *t*-test). Grafting generally reduced this parameter in both colored fruit (Figure 3E,F). The highest antioxidant capacity values were found in both the green (3.84 mg TE g^−1^ FW) and red (11.1 mg TE g^−1^ FW) fruit produced by the “Valencia” plants, followed by “Sueca,” whereas the lowest antioxidant capacity was recorded in fruit from “Cuerno.” A significant grafting and landrace interaction was observed only for the red fruit, which became evident as the highest antioxidant capacity was reported in the fruits produced by grafted and ungrafted “Valencia” plants, and by the ungrafted “Sueca” plants (Figure 3F). The lowest antioxidant capacity values were detected in the fruits from the grafted “Cuerno” plants.

The most evident result was the lower Chl a and b content in the red vs. the green fruit (*p* ≤ 0.001; Student’s *t*-test) (Figure 4A,B). In the green fruit, this concentration was not influenced by grafting, but was strongly impacted by landrace, with the fruit from “Cuerno” presenting the highest content, followed by “Valencia” and finally by “Sueca.” The interaction between the two variability factors was evidenced as the highest Chl a and b content was detected in the ungrafted “Cuerno” fruit, followed by the grafted “Cuerno,” “Valencia,” and ungrafted “Valencia” (Figure 4A). In the red fruit, no variability factor induced significant differences in Chl a and b content (Figure 4B). Not even did carotenoid content change in the green and red fruit because of grafting, but marked differences were found among landraces: the green fruits produced by “Cuerno” and “Valencia” had a higher carotenoid content, with both “Cuerno” together with “Sueca” in the red fruit (Figure 4C,D).

As expected, the lycopene concentration was higher in the red versus the green fruit. The mean value, independently of grafting and genotype, was 2.78 mg g^−1^ FW compared to one of 16.39 mg g^−1^ FW in the red fruit (*p* ≤ 0.001, Student’s *t*-test; Figure 4E,F). Although in lycopene content the green fruit was low, significant differences were found for grafting and landrace, but not for their interaction (Figure 4E). Grafting generally improved lycopene content (4.3- and 1.4-fold higher in the green and red fruit from the grafted vs. ungrafted plants, respectively), and the landraces that produced the fruit with the highest lycopene content in both fruits was grafted “Cuerno” (Figure 4E,F). For the red fruit, the grafting and landrace interaction was significant. The analysis underlined that grafted “Cuerno” produced the red fruit with the highest lycopene content, followed by grafted “Sueca” (Figure 4F). both grafted and ungrafted “Valencia” were the landraces that produced red fruit with the lowest lycopene content.

To understand the contribution of the different phytochemicals in both colored peppers, a correlation analysis was carried out between the different compounds and antioxidant capacity. In the green fruit from the ungrafted or grafted plants, a positive correlation appeared only between antioxidant capacity and phenols, even when the grafting technique was not considered (Table 2). The correlation found for the red fruit between phenols and antioxidant capacity was significant, but only for the grafted plants. For the red fruit, the correlation was significant when grafting was not considered. Chlorophyll and carotenoid content did not correlate with antioxidant capacity in the green and red fruit, whereas lycopene content correlated inversely with antioxidant capacity in the green fruit when grafting was not contemplated, and in the red fruit (Table 2). The correlations were also determined for each landrace, independently of the grafting technique, in both colored peppers (Table 3). In the “Cuerno” fruit, a significant correlation was found only in the red fruit between phenols and antioxidant capacity. Once again chlorophyll and carotenoid contents in the green and red fruit did not correlate with antioxidant capacity in any landraces, whereas lycopene was negatively related.

### 3.4. Volatile Compounds

The pepper volatile profile was characterized by 51 compounds, identified in line with both mass spectra and linear retention index (Table 4). VOCs were classified according to their corresponding chemical class and relative distributions and were included in the following nine groups: organic acids, alcohols, aldehydes, alkanes, ketones, terpenes, esters, aromatic hydrocarbons, and furans. The nine compounds belonging to the different and fewer representative classes were grouped as “miscellaneous components.” The largest portion of VOCs present in peppers was represented by terpenes (42%), followed by aldehydes (16%), esters (11%), and alcohols (6%). Ripening level was the factor that most influenced the VOC profile. As reported in Table 3, most VOCs were significantly higher in the green fruit, except for hexanoic acid, 2-ethylhexanoic acid, decanoic acid, n-dodecane, 3-methyltridecane, 4-methyl 2-pentanone, cis-tagetone, beta-trans-ocimene, acetic acid ethyl ester, n octhyl formate, o-xylene, mesithylene, N-methylpyrrole or 1-methylpyrrole and carbon disulfide, which showed a non-significant difference. On the contrary, only a few VOCs showed significant differences in the three landraces: “Cuerno” had a higher level of ethanol and a lower level of β-linalool and 5-5 methyl 1,3 dithian-2-one, while “Valencia” had a lower level of nonenal, α-santalene, salycilic acid and ethyl ester, and higher 2-heptanone, cyclobutene, and furan 2,3-dihydro-4-methyl contents.

A multivariate discriminant analysis was performed to further investigate whether a VOC’s signature discriminated the grafted from the ungrafted peppers. Of the 51 VOCs initially detected in peppers, and for both ripening levels, 10 VOCs were retained at the end of the SDA (*p* < 0.001): 4-methyl-2,3-dihydrofuran, allo-ocimene, nonanal, 2,4,6-trimethylanisole-, cyclosativene, methyl salicylate, octanoic acid, nonanoic acid, ethyl hexadecanoate for the green fruit and carbon disulfide, ethanol, pentanal, 2-propyldiene-1-cyclobutene, 2-heptanone, 7-methyl-1-octene, 2-Isobutyl-3-methoxypyrazine, beta-linalool, 5,5-dimethyl-1,3-dithian-2-one, salicylic acid methyl ester for the red fruit. The selected variables had a high discriminant power, with R^2^ ranging from 0.57 to 0.30. 

The CDA was applied to the selected variables which gave significant (*p* < 0.01) new variables called canonical variables (two canonical variables for the green fruit and one for the red fruit). For the green fruit, the first two variables accounted for 98% of total variability (Table 5), which thus indicates that the multivariate structure of the VOC could be well represented by only the first two canonical variables. This was confirmed by the scatter plot of canonical_1 × canonical_2 (Figure 5), which allowed the segregation of the six groups. Instead, for the red fruit, only the first canonical variable represented the multivariate structure by explaining 96% of variance (Table 5). Figure 6 demonstrates the discrimination effect of this canonical variable. For the green fruit, the canonical_1 variable markedly separated landraces. This separation was particularly evident for the grafted landraces (“Cuerno” scored higher than “Valencia,” while “Sueca” was intermediate between both). The ungrafted ones were almost indistinguishable according to canonical_1, with a short, but significant, distance between them (Figure 5). The original variables, which accounted mostly for this discrimination, were allo-ocimene and cyclosativene (Table 5). The first was associated more with “Valencia”, while the second one was associated with “Cuerno” as they showed a high negative and positive correlation with canonical_1. On the contrary, canonical_2 markedly separated the grafted plants from the ungrafted ones (Figure 6). The original variables that accounted mostly for this discrimination were4-methyl-2,3-dihydrofuran, N-methylpyrrole, allo-ocimene, 2,4,6-trimethylanisole, cyclosativene, methyl salicylate, octanoic acid and ethyl hexadecanoate (Table 5). All these VOCs correlated positively with canonical_2, which demonstrates a close relation with the grafted plants.

For the red fruit, the first canonical could discriminate the six groups, which demonstrates that it was able to explain the effect of landrace and grafted effect (Figure 6). Finally, the DA classified each observation (peppers) to the correct group for both green and red fruit with 100% accuracy.

## 4. Discussion

Use of grafting is a good tool to improve vegetable production in areas where biotic-abiotic stresses can seriously determine reductions in not only crop yield, but also in production quality [50,51,52]. 

In this way, most research aims to assess the rootstock and scion interaction and the effect that this interaction has on both agronomic performance and stress resistance [10,14,53]. However, in recent years, many works report the effect of grafting on the production of vegetable quality, particularly the physico-chemical, nutritional and nutraceutical properties of the fruit produced by grafted plants [10,18,29,30]. Fruit quality is a set of attributes that determine consumer choice, which can be divided into external fruit aspect, nutritional and nutraceutical characteristics [54]. These last properties have been assumed to be of particular importance for consumers given the positive relations between fruit/vegetables and human health [55,56]. Many studies report that a healthy diet rich in fruit and vegetables, named functional food, helps to delay senescence processes in humans, but to also reduce the risk of important pathologies including cancer and cardiovascular diseases [56,57,58]. The important characteristic of functional food is its richness in antioxidant compounds, capable of scavenging ROS based on cellular oxidative stress [25].

Sweet pepper fruit falls in the category of functional foods because it is rich in ascorbic acid, carotene, phenols, and also in capsaicinoids, xanthophylls, and flavonoids and is, for this reason, characterized by high antioxidant capacity [27,28]. However, the amount of phytochemicals in peppers depends on many factors, including rootstock, although literature reports opposite results about the influence of grafting on the amount of phytochemicals in peppers. 

Our study evidenced for all three landraces that grafting positively influenced marketable yield, whereas no differences were found among the three landraces. The higher yield in the sweet peppers from the grafted plants has already been demonstrated [14,59] and is attributable to an increase in the average number of fruit per plant. This increase can be related to the more vigorous root system of rootstock NIBER^®^ that, in turn, induces an increase in water and nutrient uptake [31,32]. In addition, non-marketable yields and fruit with BER markedly dropped because grafting further increased production due to fewer losses. The minor incidence of fruit with BER symptoms caused by the grafted plants could be attributed to the rootstock better resisting abiotic stress, as previously reported in this species by Johkan et al. [60], Sanchez–Rodriguez et al. [61] and Penella et al. [13,33].

In addition to increased fruit production, the physico-chemical characteristics remained the same, or even improved, for the studied rootstock. Fruit dry matter was similar in the green fruit, or increased in the red fruit, of the three different pepper landraces grafted onto NIBER^®^ compared to the fruit from the ungrafted plants, whereas no changes were recorded for pulp thickness. Other authors have found similar results [19], showing that an increment in dry matter could be attributed to the higher uptake and transport of mineral elements [62,63].

The perception of pepper fruit sourness is related to titratable acidity. Grafting also increased titratable acidity in both red and green fruit, which is an important factor for fruit quality if we consider the role played by acids in the flavor of fresh pepper products. 

Much interest in pepper fruit quality has been shown as regards to phytochemicals, including ascorbic acid, lycopene, and phenols. However, the effect of grafting on nutraceutical aspects of pepper fruit is related to landraces, but also to the fruit maturation stage. First of all, phenolic compounds significantly decreased in both the green and red fruit obtained from the grafted plants, which has already been reported by other authors in this species and in other vegetables [18,64,65]. Phenol compounds were higher in the red than green fruits at a mean concentration of about 10.9 and 6.5 mg g^−1^ FW, respectively. These values are higher than those reported by Zhuang et al. [28] in red fruit, but are similar to those found by Blanco–Ríos et al. [66] and Chavez–Mendoza et al. [18]. Finally, the most relevant result for phenol compounds was the higher content found in fruit in relation to the ripeness state with red fruit containing significantly more phenols than green fruit, as previously reported by Lee et al. [67] and Chavez–Mendoza et al. [30].

The other important nutraceutical molecule in peppers is vitamin C, with fresh peppers being an excellent source of this compound. A different behavior was detected in both green and red fruit. The green ones produced by the grafted plants had a higher ascorbic acid content, which significantly lowered in the red ones. The ascorbic acid results reported in the literature also differ. Sanchez–Torres et al. [19] found in peppers that the amount of ascorbic acid changed depending on landraces more than on grafting, or even on both the scion/rootstock combination, as reported by López–Marín et al. [65], and a significant increase was observed by Chavez–Mendoza et al. [18]. This increase seems to be connected to the connection between scion and rootstock, which induces a good flow of water and minerals to thus improve the photosynthetic process, as reported by San Bautista et al. [68] in melon.

In our study, vitamin C was closely related to the genotype. Indeed “Sueca” and “Valencia” produced green and red fruit with higher vitamin C contents. Another interesting result, and one similar to that recorded for phenol compounds, was the amount of ascorbic acid recorded in the red fruit. Regardless of the grafting technique, vitamin C was much higher in red than in the green fruit, evidencing the important of the maturity stage. 

The drop in vitamin C in the red fruit from the grafted plants was, however, compensated by the marked increase of lycopene observed in fruit from the grafted plants. It is well-known that ascorbic acid is a sensitive compound to heat treatment, unlike lycopene. Lycopene is a strong antioxidant [69] and its concentration was higher in the red than the green fruit and grafting positively impacted its concentration on both fruit. Another major difference was that related to the genotype with “Cuerno” being the landrace that, depending on grafting, produced fruit with a high lycopene content. Previous works have reported that lycopene content depends on genotype more than on grafting [70]. Obviously, red fruit contained 4- to 5-fold more lycopene than green ones, with similar values to those found by Chavez–Mendoza et al. [30]. The results obtained about lycopene were relevant when considering that lycopene is apparently the most efficient quencher of singlet oxygen and free radicals among carotenoids [71]. In addition, it does not convert into vitamin A (as β-carotene does) and is utilized entirely as an antioxidant. It is also well-known that lycopene is not sensitive to heat treatment as, for example, ascorbic acid is, and it remains in peppers even after cooking.

In a general way, total chlorophyll content was lower in the red than in the green fruit and concomitantly increased carotenoid. This obviously took place when considering the pepper fruit maturity stage. Clearly, in the red fruit, the conversion of chloroplast into chromoplast is related to both chlorophyll degradation and the increase in carotenoid biosynthesis, as already in pepper formerly by other authors [72,73]. 

Grafting led to no changes in chlorophyll and carotenoid contents, despite the significant interaction found for the total chlorophyll amount in the green fruit and the carotenoid levels in the red fruit. The highest total chlorophyll level was recorded in the green fruit produced by the ungrafted “Cuerno” compared to the other thesis, as with the fruit produced by the grafted plants of this landrace. Conversely, for “Sueca” and “Valencia”, no differences were observed in the total chlorophyll between the fruit produced by the grafted and ungrafted plants. 

Antioxidant capacity was negatively influenced by grafting, and was higher in both colored peppers produced by the ungrafted plants, whereas a strong influence of genotype was recorded with landrace “Valencia”, which produced fruit with the highest antioxidant capacity (both green and red). These results evidenced that grafting did not improve fruit antioxidant capacity as reported by other authors [74] in tomato cultivar Cecilia F1, grafted onto He-man and Spirit. Our results contrast with those reported by Chavez–Mendoza et al. [18] in fruit from grafted bell peppers. What our results evidence is that red fruit, independently of landrace or grafting, had a higher antioxidant capacity compared to green ones in accordance with the data reported by Blanco–Ríos et al. [66] in the red fruit of cultivar “Mazurca.”

The correlation analysis revealed that antioxidant capacity was related only to the phenols compound in the green fruit of all the grafted landraces, but another picture appeared for the red fruit. In fact, a significant correlation between antioxidant capacity and phenols was found only for the grafted plants, but a significant contribution to antioxidant capacity was made in the red fruitby ascorbic acid. In the red fruits, an inverse correlation between antioxidant capacity and lycopene was noted as reported by Chavez–Mendoza et al. [30].

The analytical technique to analyze VOCs identified 51 different compounds. The terpenoid class includes a wide diversity of compounds with significant differences between fruit stage maturity and landraces. Most terpenoids are mono- and sesquiterpenes, of which 3-carene and β-linaool are mainly responsible for sweetness and fresh flowery aroma, and are typical in the fruit of peppers [75] and other fruit, vegetables and food aromas [76]. These compounds were significantly present in the green fruit, with no differences among landraces for 3-carene, but a higher concentration was observed in “Valencia” and “Sueca” compared to “Cuerno.” Among sesquiterpenes, large amounts of copaene, α-bergamotene, and β-farnesene appear. β-farnesene is involved in sweet pepper fruit flavor andappeared in large amounts in the green fruit, but with no differences among landraces. 

Even aldehydes, ketone, alcohols, and acids were recorded in pepper fruit and, once again, with the highest level in the green fruit as compared to the red ones. Within acids, nonanoic acid is prevalent, nonenal was the prevalent aldehyde and 2-octen-1-ol (E) and *cis* -3 nonel -1- ol were the prevalent alcohols. Many of these compounds derive from lipids such as linolenic and linoleic acid through the action of the lipoxygenase pathway, as reported by Luning et al. [77]. No differences among landraces were found for acids, but differences were noted for ethanol (higher in “Cuerno”) among alcohols, for aldehyde nonenal (higher in “Cuerno” and “Sueca”), and also for ketones 2-heptanone (higher in “Valencia”) and 5,5-dimethyl-1,3-dithian-2-one, which was higher in “Cuerno” and “Sueca”. As formerly reported, numerous esters are found in pepper fruit [78] that typically improve fruity flavor and floral aroma in different *Capsicum* species. Methyl-salicylate is the most abundant one in green pepper fruit, and is a compound with a strong mint odor that abounds in pepper fruit [78], and is a distinctive flavor-active volatile [79].

Among furans and aromatic hydrocarbons, 2-pentylfuran, and 2,4,6-trimethylanisole are found in larger amounts in green fruit. The first furanic compound is abundant in bell pepper fruits.

Finally, in miscellaneous compounds 2-isobutyl-3-methoxypyrazine, a potent bell pepper odorant [77], was abundant in the green fruit with no differences among landraces.

A multivariate statistical analysis was performed to make an in-depth evaluation of the differences attributable to grafting and/or landraces. For the green fruit, the CDA can be explained by two canonical variables that explained 98% of the total variability in the dataset obtained from the analysis. The loading plots from the green fruit were composed of the axes of canonical_1 with canonical_2. The grafted “Cuerno” was evidently separate from other samples by canonical_1 on the score plot, and the compound that accounted for discrimination was cyclosativene, while the discrimination of the grafted “Valencia” was related to allo-ocimene. Cyclosativene is a tetracyclic sesquiterpene whose antioxidant capacity mitigates oxidative injuries in the neurodegenerative disorders field [80]. Allo-ocimene is a monoterpen that occurs in citrus and many other essential oils, and is characterized by a harsh terpene-like and somewhat citrusy character that is also typical in bell peppers [81]. The ungrafted landraces were indistinguishable and spared by the grafted plants according to canonical_2. As shown in Table 4, the graft involved a higher content of 4-methyl-2,3-dihydrofuran, N-methylpyrrole, 2,4,6-trimethylanisole, methyl salicylate, octanoic acid, ethyl hexadecanoate, cyclosativene and allo-ocimene.

For the red fruit, only the first canonical_1 variable explained 96% of variability (Table 4). According to this discrimination, three groups were observed, characterized by groups of peppers with a short, but significant, distance among them: “Valencia” (grafted and ungrafted) and “Sueca” ungrafted obtained the highest canonical scores; the ungrafted “Cuerno” obtained negative scores; “Cuerno” and “Sueca” had intermediate score values. Unlike the green fruit, a different response of the three landraces to grafting was observed as regards the profile of VOCs. As Figure 6 depicts, “Sueca” and “Valencia” were very close when ungrafted, while the graft, “Sueca” showed a similar profile to “Cuerno”. The VOCs most associated with ”Valencia” (grafted and ungrafted) and “Sueca” were 2-propyldiene-1-cyclobutene, 2-heptanone, beta-linalool, and methyl salicylate, while ethanol characterized mainly “Cuerno” ungrafted.

## 5. Conclusions

The findings reported herein support our hypothesis that grafting could be an interesting technique to improve the yield and marketable quality, mainly BER incidence, of pepper landraces without negatively affecting intrinsic fruit quality. These results also provide useful information about the pepper grafting effect on bioactive compounds and antioxidant capacity, with variation being much more related to maturity stage and genotype. The red fruit displayed higher antioxidant capacity than the green fruit, but grafting improved only lycopene content in both red and green fruit.

The volatile fraction of peppers was composed of different chemical classes, which were linked to and influenced by the graft as the discriminant analysis demonstrated. This study may also represent a useful approach to characterize the effect of graft on plant metabolism. This effect was more evident in the green fruit than in the red ones.

## Figures and Tables

**Figure 1 antioxidants-09-00501-f001:**
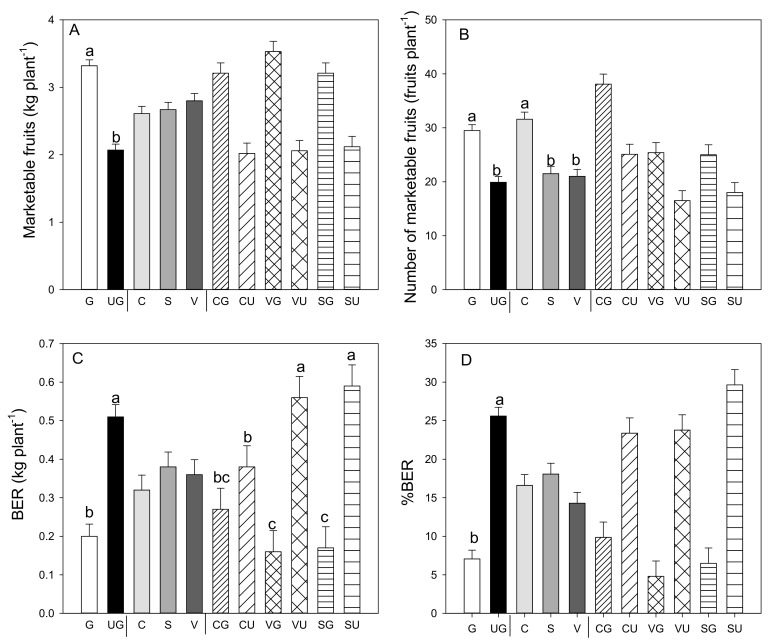
Marketable fruit yield (**A**), number of marketable fruits (**B**), production (**C**) and percentage (**D**) of fruit affected by blossom-end rot (BER) in the three pepper landraces “Cuerno” (C), “Valencia” (V), and “Sueca” (S) grafted (G) or ungrafted (U) on rootstock F1 NIBER^®^. Values are the mean ± SE of 15 plants per replicate (3 replicates) per landrace. Means were subjected to a two-way ANOVA with grafting and landrace as sources of variability. Different letters for the factors grafting and landrace, or their interaction, indicate significant differences at *p* < 0.05 using the LSD test. No letter indicates the non-significance of the *F* ratio.

**Figure 2 antioxidants-09-00501-f002:**
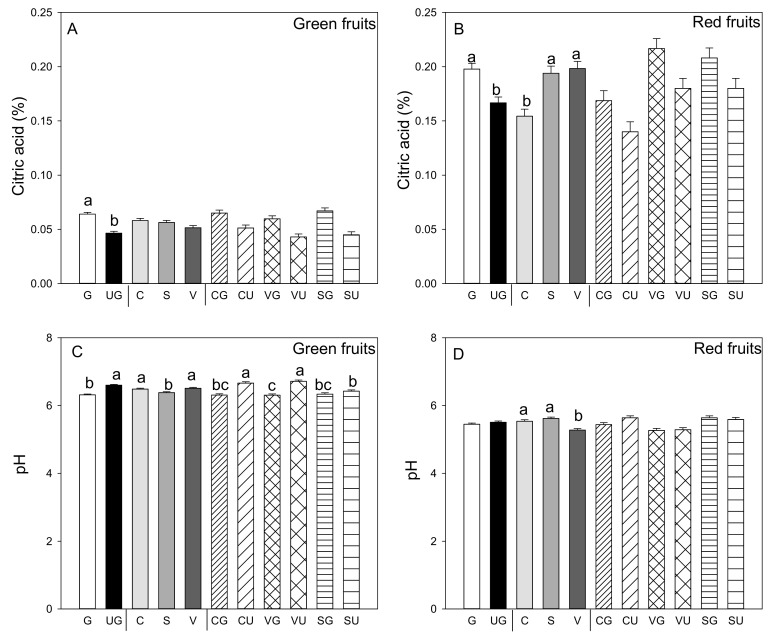
Titratable acidity (**A**,**B**) and pH values of pepper juice (**C**,**D**) in the green (**A**,**C**) and red (**B**,**D**) fruit produced by pepper landraces “Cuerno” (C), “Valencia” (V), and “Sueca” (S) grafted (G) or ungrafted (U) onto rootstock F1 NIBER^®^. Values are the mean ± SE of three fruits per replicate (3 replicates) per landrace. Means were subjected to a two-way ANOVA with grafting and landrace as sources of variability. Different letters for the factors grafting and landrace, or their interaction, indicate significant differences at *p* < 0.05 using the LSD test. No letter indicates the non-significance of the *F* ratio.

**Figure 3 antioxidants-09-00501-f003:**
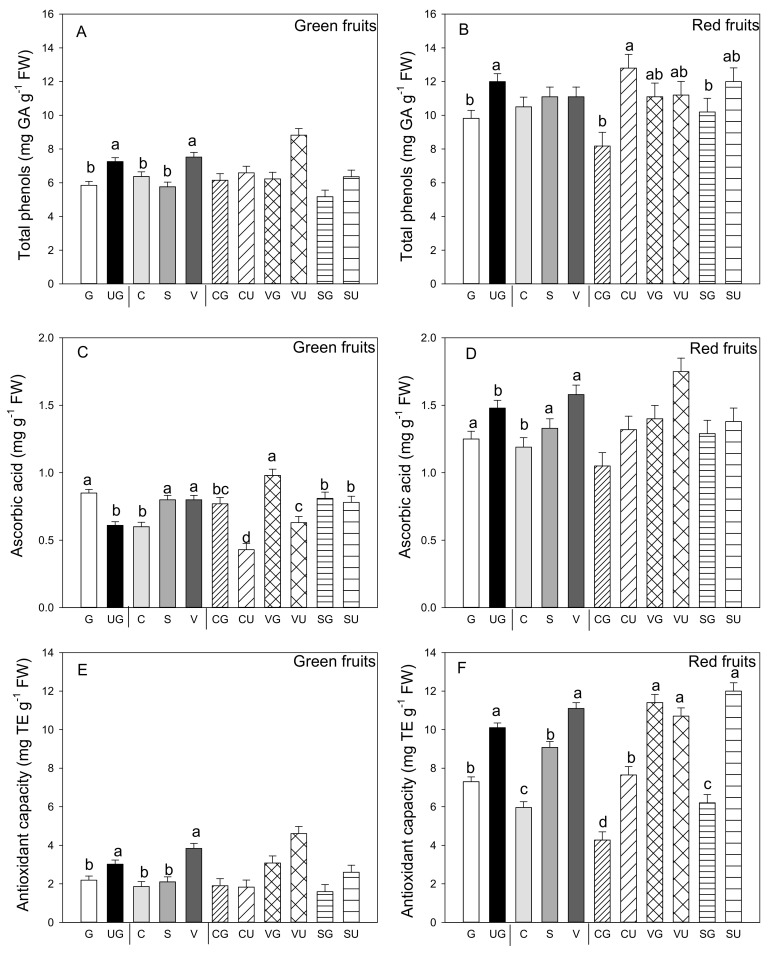
Total phenols (**A**,**B**), ascorbic acid amount (**C**,**D**) and antioxidant capacity (**E**,**F**) in the green (**A**,**C**,**E**) and red (**B**,**D**,**F**) fruit produced by the three pepper landraces “Cuerno” (C), “Valencia” (V), and “Sueca” (S) grafted (G) or ungrafted (U) on rootstock F1 NIBER^®^. Values are the mean ± SE of three fruits per replicate (3 replicates) per landrace. Means were subjected to a two-way ANOVA with grafting and landrace as sources of variability. Different letters for factors grafting and landrace, or their interaction, indicate significant differences at *p* < 0.05 using the LSD test. No letter indicates the non-significance of the F ratio.

**Figure 4 antioxidants-09-00501-f004:**
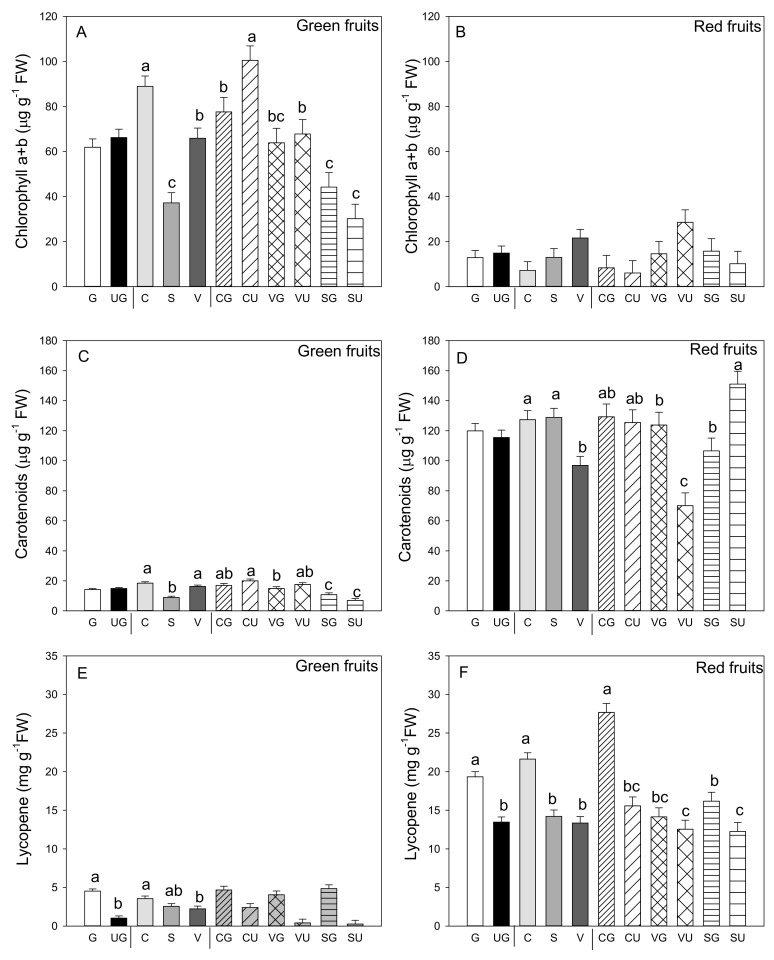
Chlorophyll a and b (**A**,**B**), carotenoid (**C**,**D**) and lycopene amount (**E**,**F**) in the green (**A**,**C**,**E**) and red (**B**,**D**,**F**) fruit produced by the three pepper landraces “Cuerno” (C), “Valencia” (V), and “Sueca” (S) grafted (G) or ungrafted (U) onto rootstock F1 NIBER^®^. Values were the mean ± SE of three fruit per replicate (3 replicates) per landrace. Means were subjected to a two-way ANOVA with grafting and landrace as sources of variability. Different letters for factors grafting and landrace, or their interaction, indicate significant differences at *p* < 0.05 using the LSD test. No letter indicates the non-significance of the *F* ratio.

**Figure 5 antioxidants-09-00501-f005:**
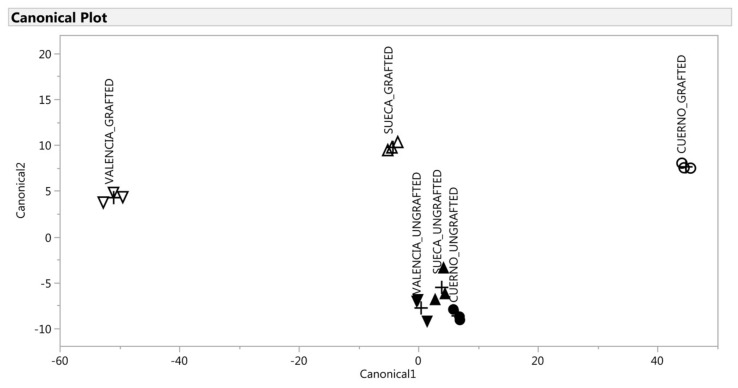
Discrimination of the VOCs of the green fruit based on the canonical discriminant analysis. ▲: SUECA_ungrafted; ∆: SUECA_grafted; ▼: “Valencia”_ungrafted; ∇: “Valencia”_grafted; ●: “Cuerno”_ungrafted; ○: “Cuerno”_grafted.

**Figure 6 antioxidants-09-00501-f006:**
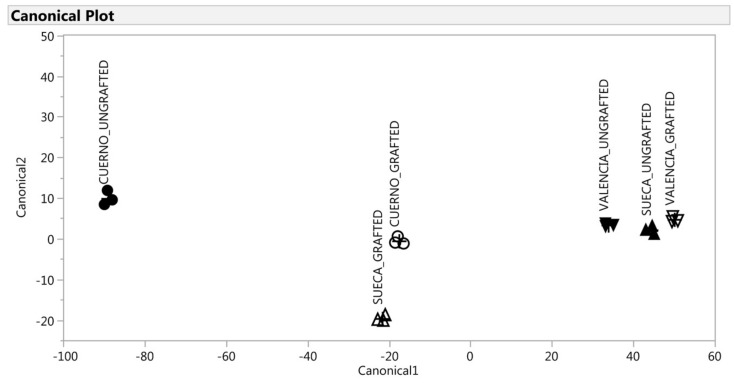
Discrimination of the VOCs of the red fruit on the basis of the canonical discriminant analysis. ▲: SUECA_ungrafted; ∆: SUECA_grafted; ▼: “Valencia”_ungrafted; ∇: “Valencia”_grafted; ●: “Cuerno”_ungrafted; ○: Cuerno_grafted.

**Table 1 antioxidants-09-00501-t001:** Dry weight and color indices in the green and red fruits produced by the three pepper landraces “Cuerno” (C), “Valencia” (V), and “Sueca” (S) grafted (G) or ungrafted (U) on rootstock F1 NIBER^®^. Values are the mean of three fruits per replicate (3 replicates) per landrace. Means were subjected to a two-way ANOVA with grafting and landrace as sources of variability. Different letters for the factors grafting and landrace, or their interaction, indicate significant differences at *p* < 0.05 using the LSD test. No letter indicates the non-significance of the *F* ratio. ns: *p* > 0.05; *: *p* ≤ 0.05; **: *p* ≤ 0.01.

	Dry Weight (%)				Hue angle (H)				Chroma (C)			
	Green		Red		Green		Red		Green		Red	
Grafting (G)												
Grafted (G)	6.45		9.12	a	110.9		31.3	b	21.7		32.0	
Ungrafted (U)	6.31		8.58	b	111.7		33.9	a	21.9		30.0	
Landrace (L)												
“Cuerno” C)	6.46		8.17	b	112.1		30.0	b	22.7		32.1	
“Sueca” (S)	6.44		9.97	a	110.4		30.4	b	22.0		29.8	
“Valencia” (V)	6.24		8.41	b	111.5		37.4	a	20.8		31.1	
G*L												
CG	6.66		8.09	d	111.1		30.3		22.7		31.2	
SG	6.43		10.4	a	110.0		29.2		21.0		29.0	
VG	6.26		8.89	c	111.7		34.3		21.4		35.8	
CU	6.26		8.26	d	113.2		29.8		22.6		33.0	
SU	6.45		9.55	b	110.7		31.6		23.0		30.6	
VU	6.22		7.92	d	111.3		40.5		20.2		26.5	
ANOVA (*df*)	% Sum of the squares											
G (1)	3.59	ns	8.76	^**^	4.80	ns	9.42	^*^	0.11	ns	5.00	ns
L (2)	7.03	ns	74.9	^**^	16.2	ns	59.9	^**^	6.63	ns	4.29	ns
G*L (2)	6.60	ns	7.57	^*^	7.83	ns	9.65	ns	5.03	ns	33.0	ns
Residuals (12)	82.8		8.76		71.2		21.0		88.2		57.8	
SD. ^(+)^	0.41		0.33		1.87		2.46		3.42		4.21	

^(+)^ Calculated as the square root of the residual sum of squares. *df*: degrees of freedom; SD: standard deviation.

**Table 2 antioxidants-09-00501-t002:** Correlation coefficients and their significance determined between each phytochemical and antioxidant capacity produced by the green and red fruit produced by the grafted (G) or ungrafted (UG) landraces on rootstock F1 NIBER^®^. Correlation coefficient r is reported when the correlation was significant. ns: *p* > 0.05; *: *p* ≤ 0.05; **: *p* ≤ 0.01; ***: *p* ≤ 0.001.

Phytochemicals	GREEN UG PLANTS	GREEN G PLANTS	GREEN FRUITS	RED UG PLANTS	RED G PLANTS	RED FRUITS
Phenols	** r = 0.85	** r = 0.77	*** r = 0.83	ns	* r = 0.71	* r = 0.53
Total chlorophylls	ns	ns	ns	ns	ns	ns
Carotenoids	ns	ns	ns	ns	ns	ns
Lycopene	ns	ns	* r = −0.57	* r = 0.68	** r = −0.78	*** r = 0.81
Ascorbic acid	ns	ns	ns	ns	* r = 0.65	** r = 0.56

**Table 3 antioxidants-09-00501-t003:** Correlation coefficients and their significance determined between each phytochemical and antioxidant capacity produced by the green and red fruit produced by the three landraces. Correlation coefficient r is reported when the correlation was significant. ns: *p* > 0.05; *: *p* ≤ 0.05; **: *p* ≤ 0.01; ***: *p* ≤ 0.001.

Phytochemicals	GREEN CUERNO	GREEN VALENCIA	GREEN SUECA	RED CUERNO	RED VALENCIA	RED SUECA
Phenols	ns	ns	*** r = 0.96	** r = 0.91	ns	ns
Total chlorophylls	ns	ns	ns	ns	ns	ns
Carotenoids	ns	** r = 0.85	ns	ns	ns	** r = 0.85
Lycopene	ns	* r = −0.83	ns	** r = 0.97	ns	ns
Ascorbic acid	ns	ns	ns	ns	ns	ns

**Table 4 antioxidants-09-00501-t004:** Effect of ripening and landrace on the volatile profile of peppers.

	Ripening Level		SEM	*p* Value	Landrace			SEM	*p* Value
	Green	Red			“Cuerno”	“Sueca”	“Valencia”		
***acids***									
hexanoic acid	0.40	0.23	0.09	ns	0.54	0.15	0.26	0.11	ns
2-ethylhexanoic acid	0.40	0.25	0.10	ns	0.46	0.12	0.37	0.12	ns
n-dodecanoic acid	1.33	0.59	0.18	**	1.34	0.79	0.74	0.22	ns
octanoic acid	1.17	0.54	0.12	***	0.97	0.66	0.95	0.14	ns
nonanoic acid	3.17	1.69	0.37	**	2.99	1.58	2.71	0.45	ns
decanoic acid	0.23	0.21	0.07	ns	0.22	0.14	0.30	0.09	ns
***alcohols***									
ethanol	2.43	0.31	0.32	***	2.64^A^	1.03^B^	0.43^B^	0.39	**
2-octen-1-ol (E)	6.10	0.41	0.48	***	3.02	3.38	3.37	0.59	ns
cis -3 nonel -1- ol	7.42	0.08	1.44	***	3.73	6.09	1.43	1.76	ns
***aldehydes***									
acetic aldheyde	0.56	0.04	0.05	***	0.36	0.29	0.25	0.06	ns
pentanal	0.25	0.09	0.04	*	0.09	0.27	0.15	0.05	ns
hexanal	2.39	0.79	0.22	***	1.37	1.78	1.61	0.27	ns
(Z)-4-heptenal	2.20	0.28	0.17	***	1.06	1.32	1.35	0.21	ns
nonanal	2.15	0.59	0.19	***	1.76	1.15	1.19	0.24	ns
nonenal	24.30	0.72	4.41	***	18.05^a^	17.47^a^	2.01^b^	5.40	*
nonadien 2-(trans)-6-(CIS)-al	7.84	2.09	1.27	**	6.41	5.14	3.34	1.56	ns
(2E,4E)-2,4-decadienal	1.49	0.07	0.12	***	0.83	0.76	0.75	0.15	ns
***alkanes***									
2-2, dimethyldecane	0.86	0.23	0.11	***	0.41	0.70	0.53	0.13	ns
n-dodecane	0.54	0.39	0.14	ns	0.28	0.54	0.57	0.17	ns
3-methyltridecane	0.13	0.01	0.06	ns	0.00	0.19	0.02	0.07	ns
1-cyclopropylpentane	1.18	0.27	0.11	***	0.71	0.72	0.76	0.13	ns
***ketones***									
1-penten-3-one	1.23	0.59	0.16	**	0.89	1.14	0.71	0.20	ns
4-methyl 2-pentanone	0.03	0.01	0.01	ns	0.01	0.00	0.05	0.02	ns
2-heptanone	0.70	0.37	0.10	*	0.31^b^	0.50^b^	0.80^a^	0.12	*
cis-tagetone	3.23	0.23	2.00	ns	5.15	0.04	0.00	2.78	ns
5,5-dimethyl-1,3-dithian-2-one	6.70	2.21	0.68	**	1.52^b^	5.66^a^	6.20^a^	0.83	***
***terpenes***									
2-propenyldiene-1-cyclobutene	0.04	0.79	0.11	***	0.07^B^	0.45^B^	0.73^A^	0.14	**
3-carene	75.05	0.13	12.61	***	26.45	39.96	46.35	15.45	ns
beta-trans-ocimene	0.11	0.05	0.05	ns	0.05	0.12	0.07	0.06	ns
7-methyl-1-octene	0.60	0.09	0.07	***	0.44	0.28	0.31	0.09	ns
Allo-ocimene	1.66	0.03	0.28	***	0.55	0.92	1.06	0.34	ns
copaene	11.95	2.69	2.10	*	7.93	10.18	3.86	2.57	ns
cyclosativene	1.46	0.48	0.25	*	1.12	1.22	0.57	0.31	ns
β-linalool	5.70	1.50	0.62	***	1.44^b^	3.79^a^	5.55^a^	0.76	**
α-santalene	0.62	0.05	0.11	***	0.57^a^	0.35^a^	0.09^b^	0.13	*
α-Bergamotene	9.53	0.57	1.69	***	7.74	5.08	2.34	2.06	ns
β-farnesene	2.28	0.39	0.34	***	1.99	1.16	0.86	0.42	ns
***esters***									
	0.34	0.16	0.07	ns	0.40	0.20	0.14	0.09	ns
n-octyl formate	0.00	0.04	0.02	ns	0.05	0.00	0.01	0.03	ns
methyl salicylate	24.80	4.83	4.28	**	12.50	21.41	10.54	5.24	ns
ethyl salicylate	1.69	0.09	0.33	**	1.72^a^	0.93^a^	0.02^b^	0.41	*
ethyl hexadecanoate	0.50	0.07	0.11	**	0.49	0.15	0.23	0.13	ns
***aromatic hydrocarbons***									
o-xylene	0.03	0.01	0.01	ns	0.01	0.02	0.03	0.01	ns
mesithylene	0.05	0.11	0.03	ns	0.05	0.13	0.06	0.04	ns
2,4,6-trimethylanisole	1.92	0.09	0.28	***	0.80	1.48	0.74	0.34	ns
***furans***	0.29	0.12	0.04	**	0.16	0.22	0.24	0.04	ns
furan,2,3-dihydro-4-methyl	0.70	0.37	0.10	*	0.31^b^	0.50^b^	0.80^a^	0.12	*
2-pentylfuran	3.19	0.32	0.39	***	1.24	2.02	2.00	0.48	ns
***miscellaneous component***									
nitrogen oxide	1.00	0.34	0.14	**	0.51	0.69	0.81	0.17	ns
N-methylpyrrole	0.40	0.32	0.07	ns	0.38	0.41	0.29	0.09	ns
2-Isobutyl-3-methoxypyrazine	37.11	3.07	3.15	***	18.84	23.01	18.42	3.86	ns
carbon disulfide	0.41	0.42	0.12	ns	0.42	0.49	0.35	0.12	ns

^a,b^ Means with different letters within the same row are statistically different (*p* < 0.05). ^A,B^ Means with different letters within the same row are statistically different (*p* < 0.01). SEM = Standard error. ns: *p* > 0.05; *: *p* ≤ 0.05; *p* ≤ 0.05; **: *p* ≤ 0.01; ***: *p* ≤ 0.001.

**Table 5 antioxidants-09-00501-t005:** Correlations between the total canonical structure and the original variables.

	Canonical_1	Canonical_2
*Green Peppers*		
4-methyl-2,3-dihydrofuran	−0.09	0.84
N-methylpyrrole	0.18	0.54
allo-ocimene	−0.40	0.47
nonanal	0.27	0.34
2,4,6-trimethylanisole	−0.03	0.64
cyclosativene	0.49	0.80
methyl salicylate	0.19	0.54
octanoic acid	0.04	0.57
nonanoic acid	0.23	0.32
ethyl hexadecanoate	0.26	0.45
*Variance explained (%)*	*91.59*	*6.59*
*Cumulative variance (%)*	*91.59*	*98.18*
*Red peppers*		
carbon disulfide	−0.083	
ethanol	−0.739	
pentanal	−0.155	
2-propenyldiene-1-cyclobutene	0.789	
2-heptanone	0.429	
7-methyl,1-octene	0.270	
2-Isobutyl-3-methoxypyrazine	0.392	
beta-linalool	0.501	
5,5-dimethyl-1,3-dithian-2-one	0.392	
salicylic acid methyl ester	0.506	
*Variance explained (%)*	*96.14*	

The correlation of VOCs associated to the canonical variables are shown in bold.

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
