# Peer review of "Effect of Grafting on the Production, Physico-Chemical Characteristics and Nutritional Quality of Fruit from Pepper Landraces"

_antioxidants, 2020, doi:10.3390/antiox9060501_

Round 1

Reviewer 1 Report

In my opinion this manuscript may be published in Antioxidants after the minor revision. Besides, the manuscript meets the criteria of scientific rigour. The Authors have used the appropriate analytical methods, and the obtained results have been properly discussed. The comments for the Authors are given below.

Page 2 line 77 „organoleptic characteristics” should be better changed as „sensory characteristics”. However, I have some doubts if  Authors can define the parameters such as: dry matter, titratable acidity, volatiles… as organoleptic (or sensory) characteristics. In my opinion they should be named  as physico-chemical  parameters (characteristics). The parameters determining sensory characteristics, such as e.g., flavour, texture, appearance, tastiness may be only evaluated by an appropriate sensory panel with proven sensory sensitivity. Thus, titratable acidity determined by the Authors can only correlate with sensory parameter, such as: acid taste (or sourness) which are estimated by the trained sensory panel. Thus, in my opinion titratable acidity cannot be define as organoleptic or sensory parameters.

Page 8 line 290, 293 and 296 “phenol” should be changed as “phenols” or “phenolics”. Phenol is hydroxybenzene (C6H5OH)

Page 12 line 381 “acids” should be changed as “organic acids”

Page 12 line 389 “pyrrole 1 methyl” should be changed as “N-methylpyrrole or 1-methylpyrrole”

Page 13 Table 4

“Nonadien 2(trans)-6(CIS)-al” should be changed as “Nonadien 2-(trans)-6-(cis)-al”

“Pentane, 1-cyclopropyl” should be changed as “1-cyclopropylpentane”

“5-5 methyl 1,3 dithian 2 one” should be changed as “ 5,5-dimethyl-1,3-dithian-2-one”

“7-methyl,1-octene” should be changed as “7-methyl-1-octene”

“Acetic acid ethyl ester” should be changed as “ ethyl acetate”

“n octhyl formate” should be changed as “ n-octyl formate”

“anisole,2,4,6-trimethyl” should be changed as “2,4,6-trimethylanisole”

“pyrrole 1 methyl” should be changed as “N-methylpyrrole”

“salicylic acid methyl ester” should be changed as “methyl salicylate”

“salicylic acid ethyl ester” should be changed as “ethyl salicylate”

Page 15 line 419 the names should be changed into “2,4,6-trimethylanisole”, “4-methyl-2,3-dihydrofuran”, “1-methylpyrrole”, (line 420) “methyl salicylate”, (line 421) “2-propyldiene-1-cyclobutene”, (line 422) “7-methyl-1-octene”, “5,5-dimethyl-1,3-dithian-2-one”

Table 5

The names of chemicals should be changed into: “ 4-methyl-2,3-dihydrofuran”, “N-methylpyrrole”, “2,4,6-trimethylanisole”, “methyl salicylate”, “2-propenyldiene-1-cyclobutene”, “5,5-dimethyl-1,3-dithian-2-one”.

Page 16 line 443 the names should be changed as “ 4-methyl-2,3-dihydrofuran”, “N-methylpyrrole”, 2,4,6-trimethylanisole” (line 444) “methyl salicylate”,

Page 19 line 582 “5-5 methyl 1,3 dithian 2” should be changed as “5,5-dimethyl-1,3-dithian-2-one” (line 587) “2,4,6-trimethylanisole” (line 589) “2-isobutyl-3-methoxypyrazine” (line 603) “4-methyl-2,3-dihydrofuran”, “N-methylpyrrole”, “2,4,6-trimethylanisole”, “methyl salicylate”

Page 20 line 613 The names should be changed as “ 2-propyldiene-1-cyclobutene”, “methyl salicylate”

Author Response

Dear reviewer 1

Reviewer 2 Report

GENERAL COMMENTS

It is a very remarkable manuscript, which is acceptable with minor revision. Nowadays, few works have been published examining the effect of grafting on phytonutrients in vegetable crops, especially in pepper and its volatile compounds.

INTRODUCTION

L84: Please replace here the explanation of the abbreviation IVIA from line 89.

MATERIALS AND METHODS

L:132: Equation of Chroma is . Please see suggested paper (Koncsek et al., 2016).

L170: Method of lycopene analytics cited from Adejo et al., (2015) is originated from Liana et al., 2009, who used method from Sharma & Le Maguer (1996), but it was detected at 502 nm. Please clarify.

RESULTS

L242: Figure 1. Some letters indicated significant differences is missing.

L259-266: Table 1: The value of the Hue angle cannot be greater than 360, because the calculation is based on trigonometry. Values of green peppers usually were below 100, values above 290 tend to be a shade of blue. Please see suggested paper (Koncsek et al., 2016) and check calculation method.

L280: Figure 2. Some letters indicated the significant differences is missing.

DISCUSSION

L460: Please insert “vegetable”; … improve vegetable production…

L464: Please insert “vegetable”; … grafting on production of vegetables quality…

L466-467: Suggested citation for the sentence, Labrie et al., (2020).

L525-526; 535-537: Temperature has opposite effect on ascorbic acid and lycopene. In the optimum temperature range of crop demand higher temperature cause more ascorbic acid and less lycopene, than lower temperature. There is inhibitory temperature effect on lycopene biosynthesis above 32 °C, while it is not justified for ascorbic acid production (Helyes et al., 2007; Pék et al., 2011).

REFERENCES

Suggested citations:

Koncsek, A., Horváth, Z. H., Véha, A., Daood, H. G., & Helyes, L. (2016). Colour evolution of conventionally and organically cultivated Hungarian red spice paprika varieties. Analecta Technica Szegedinensia, 10(1), 6-15. https://doi.org/10.14232/analecta.2016.1.6-15(http://www.analecta.hu/index.php/analecta/article/view/193/143)

Labrie, C.W., Sijtsema, S.J., Snoek, H.M., Raaijmakers, I., Aramyan, L.H. (2020). Flavour and nutrition of fruits and vegetables create added value to consumers. Acta Horticulturae, 1277, 425-432.

Reviewer 3 Report

The manuscript no. antioxidants-804976 reports very interesting results about the effect of grafting and fruit ripening on the production, organoleptic and nutritional quality of fruit from Spanish local pepper landraces.

Comments:

Introduction

The introduction correctly place the study in a broad context and highlight why it is important. Authors described well the purpose of the work and included the woking hypothesis and the aim of the study.

Materials and Methods

The section is well structured. Some details should be added. I have provided my suggestions in the pdf file.

Results

Authors provided a clear description of the experimental results.

Discussion/Conclusion

Authors discussed the results clearly in perspective of previous studies and of the working hypothesis stated. It could be of interest for the readers have in the conclusion section future research directions.

References

Journal style has been followed correctly. 

Please, see the comments on attached .pdf file
